# SemanticDepth: Fusing Semantic Segmentation and Monocular Depth Estimation for Enabling Autonomous Driving in Roads without Lane Lines

**DOI:** 10.3390/s19143224

**Published:** 2019-07-22

**Authors:** Pablo R. Palafox, Johannes Betz, Felix Nobis, Konstantin Riedl, Markus Lienkamp

**Affiliations:** Institute of Automotive Technology, Technical University of Munich, Boltzmannstr. 15, 85748 Garching bei München, Germany

**Keywords:** autonomous driving, scene understanding, Advanced Driver Assistance Systems (ADAS), fusion architecture, deep learning, computer vision, semantic segmentation, monocular depth estimation, situational awareness

## Abstract

Typically, lane departure warning systems rely on lane lines being present on the road. However, in many scenarios, e.g., secondary roads or some streets in cities, lane lines are either not present or not sufficiently well signaled. In this work, we present a vision-based method to locate a vehicle within the road when no lane lines are present using only RGB images as input. To this end, we propose to fuse together the outputs of a semantic segmentation and a monocular depth estimation architecture to reconstruct locally a semantic 3D point cloud of the viewed scene. We only retain points belonging to the road and, additionally, to any kind of fences or walls that might be present right at the sides of the road. We then compute the width of the road at a certain point on the planned trajectory and, additionally, what we denote as the fence-to-fence distance. Our system is suited to any kind of motoring scenario and is especially useful when lane lines are not present on the road or do not signal the path correctly. The additional fence-to-fence distance computation is complementary to the road’s width estimation. We quantitatively test our method on a set of images featuring streets of the city of Munich that contain a road-fence structure, so as to compare our two proposed variants, namely the road’s width and the fence-to-fence distance computation. In addition, we also validate our system qualitatively on the Stuttgart sequence of the publicly available Cityscapes dataset, where no fences or walls are present at the sides of the road, thus demonstrating that our system can be deployed in a standard city-like environment. For the benefit of the community, we make our software open source.

## 1. Introduction

Research in the field of self-driving cars has largely increased in the last few years. One of the key modules in an Autonomous Vehicle (AV) is perception. Indeed, understanding the world through sensory inputs is crucial. Current AVs mainly rely on a combination of Light Detection and Ranging (LiDAR), radar, and camera sensors to build a (semantic) 3D map of the scene. Such a sensor suite is still too expensive to be integrated on a large scale in mass production of vehicles. On the contrary, reduced versions of such a suite are already common in current Advanced Driver Assistance Systems (ADAS). A notable example are Lane Departure Warning Systems (LDWSs), which warn the driver if the vehicle drifts outside the current lane. For such a system to work, lane lines must be present on the road. However, in many scenarios, e.g., secondary roads or some streets in cities, lane markings are either not present or not sufficiently well signaled (e.g., see Figure 1a).

In this work, we present SemanticDepth, a vision-based method to locate a vehicle within the road when no lane lines are present using only RGB images as the input. To this end, we propose to fuse together the outputs of a semantic segmentation and a monocular depth estimation architecture to reconstruct locally a semantic 3D point cloud of the viewed scene. We only retain points belonging to the road and, additionally, to any kind of fences or walls that might be present right at the road’s edges. We then compute the width of the road at a certain point on the planned trajectory and, additionally, the fence-to-fence distance, which further informs about the drivable area in environments where fences or walls exist. Figure 1b visualizes an exemplary output of our method.

Our system is suited to any kind of motoring scenario and especially useful when lane lines are not present on the road or do not signal the path correctly. The additional fence-to-fence distance computation is complementary to the road’s width estimation. We quantitatively test our method on a set of images featuring streets of the city of Munich that contain a road-fence structure, so as to compare our two proposed variants, namely the road’s width and the fence-to-fence distance computation. In addition, we also validate our system qualitatively on the Stuttgart sequence from the publicly available Cityscapes dataset, where no fences/walls are present at the sides of the road, thus demonstrating that our system can be deployed in a standard city-like environment.

Even though our system can be deployed in any kind of road scenario, the initial motivation for this work was the recent Roborace [1], a motorsport competition for AVs that takes place on Formula E racetracks, such as those shown in Figure 2. In [2], it is shown how the Roborace provides the perfect environment for developing autonomous driving functions. In such an environment, lane lines are either not present on the road (Figure 2a) or not visible enough (Figure 2b) to only rely on traditional lane detection systems. This is also the case in other non-racetrack scenarios such as minor streets in cities or secondary roads.

Overall, our contributions are the following:fusing together the outputs from a semantic segmentation and a monocular depth estimation architecture in order to reconstruct a local, semantic 3D point cloud of the viewed scene,a set of simple, but effective post-processing tools to deal with the semantic 3D point cloud in order to compute relevant information about the structure of the road ahead of the AV,a labeled dataset of images featuring Roborace [1] racetracks on which to train semantic segmentation networks to discern specifically between road and fence/wall,a quantitative evaluation on a set of images featuring streets in the city of Munich recorded with a hand-held camera,and a qualitative evaluation on the Stuttgart sequence of the publicly available Cityscapes dataset, thus demonstrating that our system can be deployed in a standard city-like environment.

## 2. Related Work

Lane Departure Warning Systems (LDWS) have been largely refined and implemented in commercially available cars. However, traditional lane detection systems, either camera-based [3,4,5,6,7] or LiDAR-based [8], fail when lane lines are not present on the road. For such scenarios, in order to locate the vehicle within the road’s width, we propose to use semantic and depth cues. In the following, we revise the state of the art in the tasks of semantic segmentation and depth estimation.

### 2.1. Semantic Segmentation

Semantic segmentation is the task of segmenting an image at the pixel level, i.e., assigning a semantic class to each pixel in the input image. A semantic segmentation network must not only determine the existence of an object *car* or of an object *road* in an image, but also the boundaries of each object and their location on the image plane. In contrast to classification tasks [9,10], it is therefore necessary to obtain dense pixel-wise predictions.

Before Deep Learning (DL) became mainstream in computer vision, different approaches were used for semantic segmentation. Ensembles of decision trees that acted directly on image pixels, such as TextonForest [11], were the state-of-the-art approaches in semantic segmentation.

Convolutional Neural Networks (CNNs) [9], which have had huge success in the task of image classification, also started being leveraged for the task of semantic segmentation. Ciresan et al. [12] presented a path classification approach where the label of each pixel was predicted from raw pixel values of a square window centered on it. The intuition behind using patches was that classification networks have fully-connected layers and, therefore, they require fixed-size images.

Later on, Long et al. [13] presented their Fully Convolutional Networks (FCNs), which popularized CNN architectures for dense predictions without any fully-connected layers. This approach allowed segmentation maps to be generated for images of any size and was significantly faster compared to the previous patch-classification approach in [12]. More precisely, the FCN-8s implementation [13] achieved a 20% relative improvement compared to the previous 62.2% mean Inference-over-Union (IoU) on the Pascal VOC2012 dataset [14].

An FCN architecture is comprised of an encoder followed by a decoder, as visualized in Figure 3. The encoder is typically a pre-trained classification architecture like VGG [15] or ResNet [16], whose goal is, broadly, to extract features from the image. The decoder’s task is to project semantically the discriminative low-resolution features learned by the encoder back into the original resolution of the input image, thus producing a dense classification and effectively up-scaling the output of the encoder such that it matches the dimensions of the original image. This results in the segmentation or prediction of a class for each individual pixel in the input frame.

The great majority of the state-of-the-art semantic segmentation architectures are based on this encoder-decoder structure. Their differences usually reside in the decoder part. Approaches like DeepLabv3 [17] or PSPNet [18] achieve high IoU, but forward-pass time for these architectures is usually too high to employ them in real-time systems. In the recent ENet [19], on the contrary, efficiency was the ultimate goal, but many important sacrifices were made in the network layers at the expense of a lower classification performance, compared to other approaches.

An architecture that sits in the middle, aiming at the best possible trade-off between accuracy and efficiency, without neglecting any of them independently, is the recent ERFNet [20]. This architecture, based on a novel residual block that uses factorized convolutions, can run at several frames per second (fps) on a modern GPU, even on embedded devices, which can be mounted on a vehicle.

In this work, we employ a simple FCN-8s architecture [13], given that our focus is not on the semantic segmentation task itself, but rather on leveraging semantic information to produce a semantic 3D point cloud of the viewed scene.

### 2.2. Monocular Depth Estimation

Extracting depth information is key in autonomous driving. Outdoors, this task is commonly tackled through the use of LiDAR and radar systems, which are active sensors. Compared to passive sensors, such as RGB cameras, the former do not require the scene to be illuminated, but their high cost and sparse output give a clear advantage to inexpensive passive camera sensors. On the downside, a monocular camera only provides 2D information and does not directly measure distances. There is a broad range of different approaches tackling the problem of depth estimation, either using image pairs [21], several overlapping images taken from various viewpoints [22], temporal sequences [23], or assuming a fixed camera, static scene, and changing lighting [24].

Whereas these approaches need to have more than one input image of the view of interest, learned monocular depth estimation only requires a single image as input. In this work, we leverage the recently proposed monodepth architecture developed by Godard et al. [25] in order to estimate depth from a monocular RGB camera.

In essence, Godard proposes a fully-convolutional DNN loosely inspired by the supervised DispNet architecture presented in [26]. By posing monocular depth estimation as an image reconstruction problem, Godard et al. [25] solve for the disparity map without requiring ground truth depth. However, only minimizing a photometric loss (Godard’s first approach was to only use the left image to reconstruct the right image) can result in good quality image reconstructions but poor quality depth. They overcame this limitation by including a left-right consistency check in their fully differentiable training loss, which improved the quality of their synthesized depth images. Note that this type of consistency check is commonly used as a post-processing step in many stereo methods, e.g., [27], but Godard incorporates it directly into the network. In our work, we leverage this architecture to produce a disparity map from an input RGB image.

### 2.3. Fusing Semantic Segmentation and Monocular Depth Estimation

Several works propose DNNs that leverage semantic information to boost the performance of a monocular depth estimation architecture [28,29]. The opposite, i.e., incorporating complementary depth information into semantic segmentation frameworks, has also been studied in [30]. In [31], the authors designed a hybrid CNN that jointly estimates depth and semantic labels from a single RGB image. In our work, we do not aim at presenting a new DNN architecture for either the joint estimation of depth and semantic information [31], or for monocular depth estimation using semantic cues [28,29]. On the contrary, we propose a method to locate a vehicle within the road (therefore, also computing the road’s width) when no lane lines are present in the scene. Such an environment poses a problem for current systems that only rely on lane line detection to position the vehicle within the road. This is especially common in Formula E racetrack scenarios temporarily built on city streets, but also typical for secondary roads, minor streets in cities, or roads under construction.

To this end, we propose to create a local, semantic 3D point cloud of the viewed scene by means of fusing the output of a monocular depth estimation network with that of a semantic segmentation network. None of the works mentioned above propose such a system. They only focus on obtaining semantic and depth information from the viewed scene, but do not attempt to translate such cues into meaningful data that can then be used by a control algorithm to keep the vehicle within the road.

## 3. Approach

We present a novel vision-based method (Figure 4) to locate a vehicle within the road when no lane lines are present, thus naturally complementing traditional LDWSs which only rely on the existence of road markings. Our approach is designed to only require as input an RGB image featuring a road and, only optionally, two fences (or walls) on each side of the road, thus being suited to any kind of motoring scenario. By fusing together the outputs from both a semantic segmentation and a monocular depth estimation DNN, our proposed method is able to obtain a local, semantic 3D point cloud of the viewed scene and then extract the road’s width and the fence-to-fence distance at a certain point ahead of the camera. Note that computing the width of the road is always equivalent to locating the vehicle within the road’s width, whilst the fence-to-fence distance only informs about the real drivable limits in environments where a fence or wall is present right at the edges of the road. Both computations can help increase the vehicle’s situational awareness, generating information that can be potentially fed to the path planning and the low-level control algorithms of an autonomous vehicle.

The intuition behind this approach is the following. On the one hand, presenting a semantic segmentation architecture with an RGB image produces 2D masks of the classes the network was trained to classify. In our case, we wanted to classify each pixel as either road, fence, or background. On the other hand, a monocular depth estimation network is able to generate the disparity map corresponding to the input RGB image. By overlaying the obtained 2D masks onto the disparity map, we can selectively back-project certain regions of the disparity map (road and fences) back to 3D, implicitly obtaining a 3D segmentation of the scene. We can then remove outliers present in the 3D point clouds and, finally, compute any relevant information or geometric distances in the scene.

As previously mentioned, we identified two approaches for relevant distance calculation: on the one hand, the road’s width at a certain point ahead of the vehicle; on the other hand, the fence-to-fence distance. The latter is the result of first fitting planes to the 3D point clouds of the road and fences, looking then for the intersection of these planes with each other (road with left fence and road with right fence), and finally, computing the distance between the two intersected lines at a certain depth ahead of the car.

In the following, we present in detail the modules that make up our proposed method. We begin by describing our strategy for semantic segmentation, then briefly outline the monocular depth estimation step, and finally, we show how the output of these first two modules can be fused together to compute a semantic 3D point cloud from which relevant information about the viewed scene can be extracted.

### 3.1. Semantic Segmentation Architecture

The range of semantic segmentation architectures to choose from is large, as discussed in Section 2.1. In this work, we leverage the FCN-8s architecture [13]. We acknowledge that more advanced options exist. However, our focus is not on the semantic segmentation task itself, but rather on leveraging semantic information to produce a semantic 3D point cloud of the viewed scene.

#### 3.1.1. Creation of Our Own Dataset for Semantic Segmentation

In semantic segmentation, creating a new dataset from scratch is not common. However, after training our FCN-8s implementation on the Cityscapes dataset [32] on roads and fences (and building walls) for 100 epochs and running inference on Roborace images, we noticed there was a huge margin for improvement (Figure 5a,b), especially in the classification of the fences. Therefore, we decided to build a semantic segmentation dataset of Roborace images, since they offered a larger number of fence/wall examples than Cityscapes.

The Roborace competition provides video sequences to its participants that are recorded by the vehicle’s frontal cameras in different cities. We selected those from the Formula E events in Berlin, Montreal, and New York. Out of the available video sequences of each city, we picked one out of every four frames and collected a total of 250 frames for every city, thus obtaining the roborace750 dataset.

Table 1 shows the distribution of frames of the roborace750 dataset, which comprises a training, a validation, and a test set. Note that the validation and the test set contained visually different frames from the Berlin racetrack.

By training on roborace750 for 100 epochs, we clearly managed to outperform the model trained on Cityscapes, as can be seen qualitatively in Figure 5c,d. The creation of our own labeled dataset of Roborace frames was therefore justified by the notable improvement in the segmentation results. Note that this is not due to overfitting to Roborace frames, since in Section 4.1 we demonstrate how this same model (trained on Roborace images) generalizes to different environments, such as streets in the city of Munich, thus allowing our system to be deployed in any motoring scenario.

In this section, we described the semantic segmentation task, one of the main building blocks of our proposed method for road width computation in scenarios where road markings are not present.

### 3.2. Monocular Depth Estimation

For the task of monocular depth estimation we leveraged the work by Godard [25]. In essence, the novelty of this monocular depth estimation architecture lies in that it poses monocular depth estimation as an image reconstruction problem, solving for the disparity map without requiring ground truth depth. For a more detailed explanation, refer to Section 2.2.

A monodepth model pre-trained on Cityscapes produced relatively good results when presented with Cityscapes images. Using this same model on different images produced less optimal results, as we found in a set of preliminary experiments. Still, in the remainder of this work, this Cityscapes-trained model is used for the task of monocular depth estimation.

### 3.3. Proposed Approach

The system we propose (Figure 4) locates a vehicle within the road relative to the road’s edges in scenarios where lane lines are not present. The pipeline can be summarized in the following steps:Read the input RGB image and re-size it to 512 × 256.Apply a semantic segmentation model to make inference on the input frame and generate both a road mask and a fence mask.Produce a disparity map using a monocular depth estimation network.Generate a 3D point cloud from the disparity map using stereo vision theory [33].Use the previously computed 2D masks and obtain semantic 3D point cloud of the scene.Remove noise from the 3D point clouds.Compute the road’s width and the fence-to-fence distance at a certain point ahead of the vehicle.

#### 3.3.1. Obtaining 2D Masks of the Road and the Fences from the Input Image

After re-sizing the input image to 512 × 256 we apply our semantic segmentation model pre-trained on Roborace in order to classify every pixel into three different classes (road, fence, or background). This step outputs a 2D mask for both the road and the fences in the scene (Figure 6).

#### 3.3.2. Obtaining a Disparity Map from the Input Image

Next, we computed the disparity map of the input image by using a monodepth model pre-trained on Cityscapes by Godard [25]. Figure 7 shows the exemplary results of such a model when applied to a Cityscapes image.

#### 3.3.3. Generating a 3D Point Cloud from the Disparity Map Generated by Monodepth

In this step, we convert the disparity map obtained in the previous step into a 3D point cloud. To back-project to 3D the image plane coordinates ${\left(u,v\right)}^{T}$ of a pixel with disparity *d*, we can use the inverse of the standard pinhole camera model, and in particular the matrix form as presented in [34]:(1)$$\begin{array}{c} \left[\begin{array}{c} X^{\prime }\\ Y^{\prime }\\ Z^{\prime }\\ W \end{array}\right]=Q\left[\begin{array}{c} u\\ v\\ d\\ 1 \end{array}\right],\;where\;Q=\left[\begin{array}{cccc} 1 & 0 & 0 & -c_{x}\\ 0 & -1 & 0 & c_{y}\\ 0 & 0 & 0 & -f\\ 0 & 0 & 1/b & 0 \end{array}\right], \end{array}$$
where ${\left(X^{\prime },Y^{\prime },Z^{\prime },W\right)}^{T}$ are the homogeneous coordinates of the 3D point, *b* the baseline or distance between left and right cameras in a hypothetical stereo camera, and *f* the focal length of the cameras; $c_{x}$ and $c_{y}$ are the 2D coordinates of the principle point. Note that we apply a rotation of 180° around the *x* axis to matrix Q so that the *y* axis points upwards, the *x* axis to the right and the *z* axis correspondingly following a right-handed coordinate system. Dividing by *W* finally produces the 3D coordinates of the back-projected point in the camera frame:(2)$$\begin{array}{c} \left[\begin{array}{c} X\\ Y\\ Z \end{array}\right]=\frac{1}{W}\left[\begin{array}{c} X^{\prime }\\ Y^{\prime }\\ Z^{\prime } \end{array}\right]. \end{array}$$

On the one hand, dealing with images from the same dataset the monodepth [25] model was trained on is straightforward in theory: we simply need to use the correct baseline and intrinsic camera parameters (focal length and principle point) of the image with which we want to work. In practice, however, the generated 3D point cloud is not correctly scaled, and we have found that further tweaking of the baseline and focal length is necessary. However, the reconstructed geometry represents to some extent the underlying scene. This behavior was already reported by Godard et al. in their work [25].

On the other hand, applying a monodepth model to images with a focal length and aspect ratio different from those of the dataset the model was trained on does not produce meaningful results. Therefore, in order to apply a monodepth model pre-trained on Cityscapes on images recorded with a camera setup with a different focal length and aspect ratio than those from Cityscapes, we proceed in the following manner. First, we scale the disparity map produced by monodepth with the original width of the input image. Second, we set a virtual baseline of 1 m. Finally, we use the intrinsic parameters, i.e., focal length, *f*, and principal point, $\left(c_{x},c_{y}\right)$, of the camera that took the image and use those parameters to back-project the point to 3D by using Equations (Equation 1) and (Equation 2). In practice, we still need to tweak manually both the baseline *b* and the focal length *f* to estimate a correct scale for the generated 3D point cloud, but this is expected, as reported in [25].

#### 3.3.4. Overlaying the 2D Masks onto the 3D Point Clouds

Using the previously obtained 2D masks of the road and the fences we can select the pixels corresponding to these classes and back-project them to 3D as explained in the previous step. Figure 8 shows several views of the resulting semantic 3D point cloud featuring only the road and the fences (or building walls, in this particular case). Note that there is a significant amount of noise, which must be dealt with in order to be able to then compute relevant distances within the 3D point cloud.

#### 3.3.5. Denoising the 3D Point Clouds

We have created a small library to deal with point clouds that fits our particular needs. Among other tools, this library offers a function to compute the median absolute deviation of a given 3D point cloud along one of the three dimensions (set by the user as a parameter), then computes a penalty for each point, and finally, removes all points whose penalty value is higher than a given threshold.

Our proposed approach first applies the function described above to the 3D point cloud of the road, both in the *x* and *y* axes. By doing so, we manage to remove most of the existing noise. In the *z* axis (perpendicular to the image plane of the camera), we apply a function to remove all points that are too close to the camera, since in the first meters, there is always a certain level of distortion in the disparity map produced by monodepth.

We can further remove noise by fitting a plane to the road and then removing all points whose distance to the computed plane is greater than a given threshold. Levering the statistical denoising functions available in the recent Open3D [35] further helps to remove outlier points completely.

With regards to the fences, the denoising process is similar. Before that, however, we need to separate the original point cloud containing both fences into two different point clouds. To this end, we compute the mean of all points belonging to the original point cloud containing the fences (left and right together), whose *x* coordinate will typically be close to zero. Then, we extract all points to the left of the mean point, i.e., points whose *x* coordinate is smaller than that of the mean point, into a left fence point cloud and conversely for the right fence.

What remains now is extracting any kind of valuable information from the denoised 3D point clouds we obtain. One relevant measure that can help increase the vehicle’s situational awareness is the width of the road. The fence-to-fence distance can also provide insightful information about the structure of the road ahead in certain scenarios where fences/walls are present.

#### 3.3.6. Road’s Width Computation

In this first approach, we only use the point cloud corresponding to the road. In order to compute the road’s width, we try to find the distance between the right-most point and left-most point of the road’s 3D point cloud at a given distance ahead of the camera along its *z* axis, which is perpendicular to the image plane.

Essentially, this consists in iterating over all points situated at a certain *z* coordinate, looking then for the two points with the highest positive and negative *x* coordinates (the *x* axis roughly goes along the width of the road in most cases), and, finally, computing the euclidean 3D distance between these two points. Figure 9 exemplifies the result of this computation.

Note that repeating this same procedure for several distances may also be of interest, though it simply depends on what kind of information the control algorithms need.

This approach could suffer from poor road segmentation. To overcome this limitation, we need a more robust approach where we not only rely on the segmentation of the road, but also use the information given by the fences.

#### 3.3.7. Fence-to-Fence Distance Computation

In this approach, we not only fit a plane to the road’s point cloud, but also to that of the left and right fences. Then, we can easily compute the intersection lines between the road’s plane and the two fences’ planes, thus obtaining the equations for both intersection lines. Finally, we can compute the distance between these two lines at a given point ahead of the camera. Figure 10 exemplifies the result of this computation.

Note that by looking for the intersection of the road’s plane with the corresponding fence’s plane we ensure that computing the road’s width does not require a 100% accurate segmentation of the road, only one good enough so that a plane can be fitted to the road.

### 3.4. Datasets Used for the Evaluation

We tested our method on a series of images featuring streets of the city of Munich (Figure 11), which we will refer to as the Munich test set. Note that the scenes viewed by these images do not contain lane markings on the road. Moreover, walls or fences very close to each side of the road are present in three of the five test frames. Both these facts allowed us to test our proposed method in its two variants.

Note that we manually recovered the ground truth width of the road 10 m ahead of the camera for each frame, thus allowing us to quantitatively test the performance of our method.

## 4. Results

In this section, we present quantitative results obtained after running our system on the Munich test set (Section 3.4) and qualitative results on the Stuttgart sequence from the publicly available Cityscapes [32] dataset.

### 4.1. Results on the Munich Test Set

We used our best Roborace-trained semantic segmentation model (Section 3.1.1). As for the monocular depth estimation model, we employed a monodepth model pre-trained on the Cityscapes stereo dataset by the authors of [25]. As discussed in Section 3.3.3, recall that it is unclear how a monodepth model behaves on images taken by cameras with a different focal length and aspect ratio than those from the images on which the model was trained.

The Munich test set was recorded with the rear camera of an iPhone 8, which has a focal length of 3.99 mm and a sensor size of 1.22 μm, and produces 4032 × 3024 images. The theoretical focal length can be computed through the following equations, which transforms the camera’s focal length from mm to pixels through the sensors’ size [33]:(3)$$\begin{array}{c} f=\frac{F}{sensor\_size\_per\_pixel}, \end{array}$$
where *f* is the focal length in pixels, *F* is the focal length in mm and sensor_size_per_pixel is the width per pixel measured in mm/pixel of one of the tiny sensors that integrate the whole camera. By substituting the corresponding values for the iPhone’s rear camera, we can obtain:(4)$$\begin{array}{c} f=\frac{3.99\;\mathrm{mm}}{1.2\times 10^{-3}\;\mathrm{mm}/\mathrm{pixel}}=3270.5\;\mathrm{pixels}. \end{array}$$

Given that the input images (of size 4032 × 3024) are re-sized to 512 × 256, we must also scale the focal lengths accordingly in both *x* and *y* directions:(5)$$\begin{array}{cc} f_{x} & =512\;\mathrm{pixel}\;\cdot \;\frac{3270.5\;\mathrm{pixel}}{4032\;\mathrm{pixel}}=415.3\;\mathrm{pixel}, \end{array}$$
(6)$$\begin{array}{cc} f_{y} & =256\;\mathrm{pixel}\;\cdot \;\frac{3270.5\;\mathrm{pixel}}{3024\;\mathrm{pixel}}=276.9\;\mathrm{pixel}. \end{array}$$

In practice, by manually calibrating the camera, i.e., by using OpenCV’s calibration function on a series of already re-sized images (256×512) featuring a chessboard, we obtained:(7)$$\begin{array}{cc} f_{x} & =480.088\;\mathrm{pixel}, \end{array}$$(8)$$\begin{array}{cc} f_{y} & =322.316\;\mathrm{pixel}, \end{array}$$
which are in the same order of magnitude as the theoretical values obtained in Equations (Equation 5) and (Equation 6). Therefore, a good estimate for the focal length *f* in Equation (Equation 1) should be within these values.

Note that due to the iPhone’s camera auto-focus, all five images from the Munich test set were taken with slightly different focal lengths. To discover the correct focal length for every image in the Munich test set we tried out different values of *f* and analyzed the difference between the predicted and the ground truth width of the road. We found that the system performed better on frames Munich-1 and Munich-5 when using a focal length of 380 pixels; on the other hand, fixing *f* to 580 pixels produced better results on the rest of the frames. Table 2 shows the results obtained when applying our system 10 m ahead of the camera on the Munich test set using the correct focal lengths for each sample frame.

In Table 2 we report: (1) the error distance, **e**, between what the pipeline considers to be a depth of 10 m and where a depth of 10 m really is; (2) the ground truth width of the road, **gt width**, 10 m ahead; (3) the predicted road’s width, **rw**; (4) the fence-to-fence distance, **f2f**; (5) the absolute error between the ground truth width of the road and the predicted road’s width, **AE rw**; (6) and the absolute error between the ground truth road’s width and the fence-to-fence distance, **AE f2f**. On the last row, we present the Mean Absolute Error (MAE) across all five images from the Munich test set for both the predicted road’s width and the fence-to-fence distance.

As a qualitative example, in Figure 12, we applied our proposed system with a focal length of 580 on the frame Munich-3. We display the frame’s segmentation together with the information about the computed distances on Figure 12b and the post-processed 3D point clouds with the predicted road’s width and the fence-to-fence distance (red and green lines, respectively) in Figure 12c,d.

### 4.2. Inference Times of the Proposed Approach on the Munich Test Set

Table 3 gathers the inference times (in seconds) of the proposed system on all images from the Munich test set. We computed the total time per frame, as well as the time per task for eight different tasks, namely: (1) reading and re-sizing the frame, **t_read**; (2) segmenting the frame, **t_seg**; (3) obtaining the disparity map, **t_disp**; (4) converting the disparity map into a 3D point cloud, **t_to3D**; (5) denoising and fitting a plane to the 3D point cloud of the road, **t_road**; (6) computing the road’s width, **t_rw**; (7) denoising and fitting planes to each of the fences in the scene, **t_fences**; and (8) computing the fence-to-fence distance, **t_f2f**. Additionally, we computed the mean time for every task, as well as the percentage that it represents in the mean overall inference time. These two measures are shown on the last two rows of the table.

Note that more than half of the time needed to process a frame completely was dedicated to reading and re-sizing the frame (54.71%). The second most time-consuming task was segmenting the image (23.97%), followed by the task of denoising the road and fitting a plane to it. The least time-consuming tasks were the actual computation of the road’s width and the fence-to-fence distance. We must bear in mind, however, that denoising and fitting planes to the road and fences are actually essential parts to the computation of both these measures.

The inference times above were obtained by employing an Nvidia Titan XP GPU.

### 4.3. Qualitative Results of the Proposed Approach on the Stuttgart Sequence of the Cityscapes Dataset

We tested our system on the Stuttgart sequence of the Cityscapes dataset. In this case, we only computed the road’s width, given that there are no fences or walls to the sides of the road in this sequence. Figure 13 visualizes some qualitative results. The complete sequence can be found at https://youtu.be/0yBb6kJ3mgQ, as well as in the Appendix A. For each frame, we also rendered a frontal and a top view of the 3D point cloud on which we computed the width of the road, which we placed at the lower left and right corners of the images, respectively. On a gray stripe at the top of every frame, we printed out the estimated road’s width 10 m ahead, as well as the distance from the camera frame to the left and right ends of the road.

By studying the complete sequence, we observed that predictions were robust against occlusions of the road, e.g., by pedestrians. Given the nature of the monocular depth estimation method we used, i.e., monodepth, the correct scale cannot be recovered. Therefore, estimates might not match exactly the real dimensions of the road. Using a calibrated stereo camera could simplify and improve the process of obtaining a metrically correct 3D point cloud.

## 5. Discussion

In this section, we further discuss the results obtained by our method on the Munich test set.

### 5.1. Discussion of the Results on the Munich Test Set

Qualitatively, Figure 12 shows how both the semantic segmentation task and the depth estimation task achieved a fairly good segmentation of the image and 3D reconstruction of the scene, respectively.

The semantic segmentation model used was that trained on roborace750 (Section 3.1.1) for 100 epochs. Note how this model, despite having been trained on Roborace images, generalized to environments it had never seen before, such as the city of Munich. Not only that, but the model was also capable of performing a good segmentation despite the fact that the aspect ratio in the Munich test set was completely different from that of the roborace750 dataset (4032 × 3024 and 1200 × 1600, respectively). This justifies again the creation of our own Roborace dataset for the task of road and, particularly, fence segmentation.

Secondly, with respect to the monodepth model pre-trained on Cityscapes, we can conclude that it was able to predict reasonable depth information when presented with images that resemble those from the Cityscapes dataset, despite the fact that neither the focal length, nor the aspect ratio of the Munich test set match those of Cityscapes. A fairly extensive tweaking of the focal length and baseline in Equation (Equation 1) is however necessary to obtain a correctly scaled 3D point cloud, but this is inherent to the monodepth model presented in [25].

An important conclusion we can draw from Table 2 is that computing the road’s width is a reliable method to locate the camera within the road when no lane lines are available, with an overall MAE of 0.48 m.

Furthermore, in environments where fences of walls are directly at the road’s edges (frames Munich-2, Munich-3 and Munich-5), computing the fence-to-fence distance provides an exceptional additional measure of the width of the drivable area ahead, with a MAE of 0.69 m for these three frames. For frames Munich-1 and Munich-4, the fence-to-fence distance was less informative, but still provided an extra layer of knowledge about the viewed scene.

Indeed, we considered that jointly computing the road’s width and the fence-to-fence distance provides a more robust understanding of the scene. Either approach could suffer from a range of different problems at any time. By considering the output of both approaches, the decisions the vehicle would ultimately have to make would be better informed.

The hypothetical problems we are referring to are essentially rooted in imperfect segmentation of the road or sub-optimal depth estimation of the scene geometry. The latter issue could be solved by directly employing a stereo rig instead of a neural network for the task of depth estimation.

### 5.2. Discussion of the Inference Times on the Munich Test Set

On the Munich test set, our method took an average of 0.6375 s to process a single image. Note that the time required for reading and re-sizing the input image accounted for more than half of the total inference time. This was to some extent expected, given that images in the Munich test set had to be re-sized from an initial resolution of 4032 × 3024 to 512 × 256.

When applying our system on Roborace images, this reading and re-sizing time fell to approximately 0.07 s, which means that the total inference time for Roborace images would be around 0.36 s, or roughly 3 fps. Our system is thus not real time, but we believe that only minor improvements would be required to achieve online capability.

With regards to the second most time-consuming task, segmentation, a time of 0.1528 s on 512 × 256 images using an FCN-8s [13] architecture was completely expected. Without considering the time required for reading and re-sizing, segmentation was the bottleneck in the pipeline. Future work would have to first address this issue, essentially by employing a faster architecture.

The third-most time-consuming task corresponded to denoising and fitting a plane to the raw 3D point cloud of the road, which accounted for roughly 13% of the total inference time. Essentially, this was due to the large number of points that belonged to the 3D point cloud of the road in comparison to the number of points that belonged to the left and right fences. For the latter, i.e., the fences, we only needed an average of 0.0146 s to denoise the point clouds and fit planes to them. Further reducing these times could be achieved by sub-sampling the point clouds.

All in all, the proposed approach is able to process a 4032 × 3024 image in roughly 0.64 s and a 1200 × 1600 image (Roborace images) in approximately 0.36 s.

## 6. Outlook

In the semantic segmentation task, the first clear step towards improving results is enlarging our own semantic segmentation dataset of Roborace images. We also believe it would be of great interest to train different semantic segmentation architectures on our Roborace dataset (or on a larger version of it) and compare the results with the ones already obtained in this work by using an FCN-8s architecture.

With regards to the task of camera-based depth estimation, it would be of interest to explore the use of stereo pairs instead of DNNs for monocular depth estimation, which would allow directly obtaining metrically correct 3D point clouds. Given the modularity of our approach, this could be easily accomplished.

Finally, running segmentation and depth estimation in two separate parallel threads would increase efficiency; in the current version of our system, we first run semantic segmentation, then depth estimation, and finally, merge the results from each module. By running both processes simultaneously, the speed of the pipeline –now around 0.36 s for a 1200×1600 frame– could be reduced notably.

## 7. Conclusions

In this work, we presented a vision-based method to estimate the width of the road (thus, also locating the vehicle within the road) in scenarios where lane lines are either not present or not visible enough to rely on traditional LDWS. This was accomplished by fusing together the outputs of a semantic segmentation and a monocular depth estimation architecture to build a local, semantic 3D point cloud of the viewed scene on which we could then: (1) compute the road’s width by looking for the right-most and left-most 3D points of the road at a certain point ahead of the vehicle; and (2) compute the fence-to-fence distance by first fitting planes to the 3D point clouds of the road and the fences, then finding the intersection between these planes and, finally, computing the distance between the intersected lines at a certain point ahead.

We successfully tested our method in its two variants on a series of images featuring streets of Munich, obtaining robust and accurate predictions close to the ground truth width of the road. We also evaluated our system qualitatively on the Stuttgart sequence of the Cityscapes dataset.

Minor improvements, mainly targeted towards enabling real-time capability, would allow our proposed system to be integrated into the autonomy stack of AVs, contributing to increasing their situational awareness in general and in scenarios where no lane lines are present in particular.

## Figures and Tables

**Figure 1 sensors-19-03224-f001:**
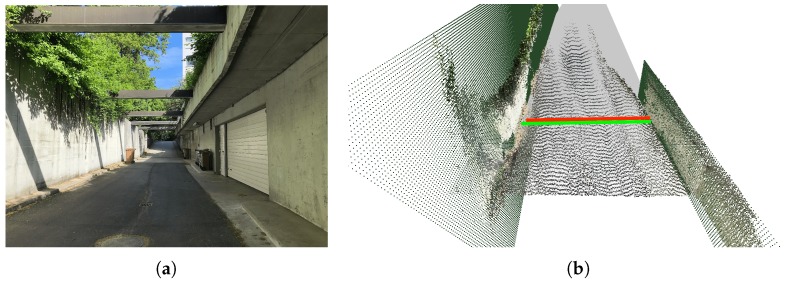
Our method locates the vehicle within the road in scenarios where no lane lines are available. To do so, we reconstruct a local, semantic 3D point cloud of the viewed scene and then propose two complementary computations: (1) extracting the width of the road by employing only the road’s 3D point cloud and (2) additionally leveraging fences/walls to the sides of the road to compute the fence-to-fence distance. Our system can work in any kind of motoring scenario. (**a**) Original image. (**b**) Computation of the road’s width and the fence-to-fence distance.

**Figure 2 sensors-19-03224-f002:**
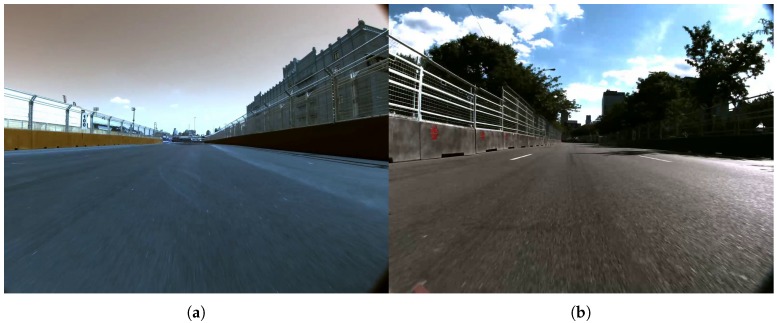
Example views of Roborace racetracks. The main elements are the road and the fences. (**a**) Frame from New York’s racetrack. (**b**) Frame from Montreal’s racetrack.

**Figure 3 sensors-19-03224-f003:**
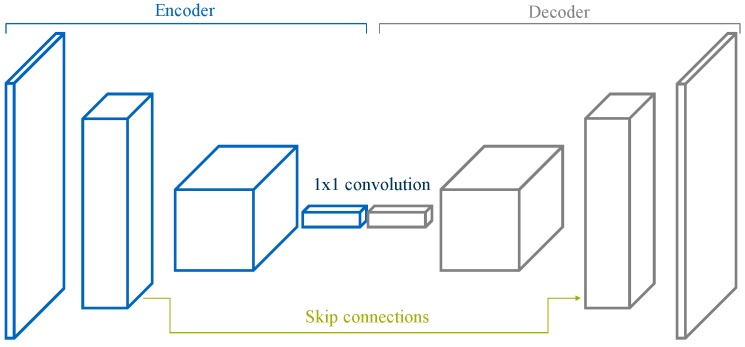
Typical scheme of a fully-convolutional network [13].

**Figure 4 sensors-19-03224-f004:**
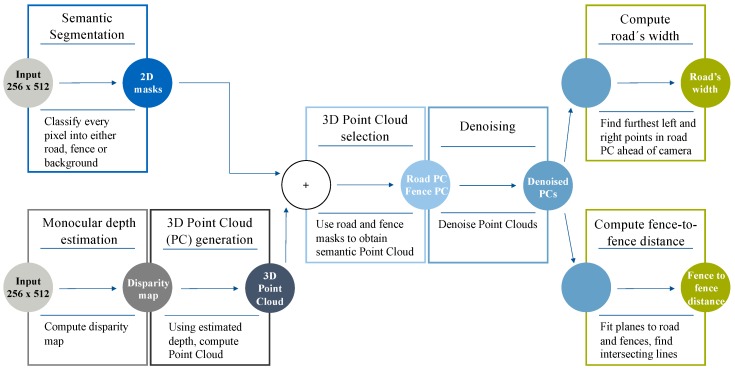
Pipeline describing our proposed approach.

**Figure 5 sensors-19-03224-f005:**
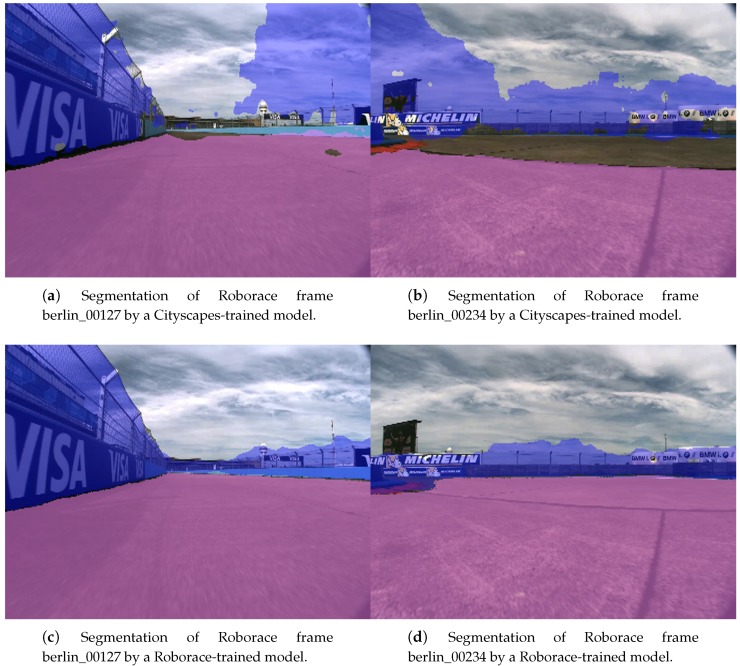
Segmentation of Roborace frames berlin_00127 (left) and berlin_00234 (right) into the classes road (purple) and fence (blue) by a model trained on Cityscapes (**a**,**b**) and by a model trained on Roborace images (**c**,**d**).

**Figure 6 sensors-19-03224-f006:**
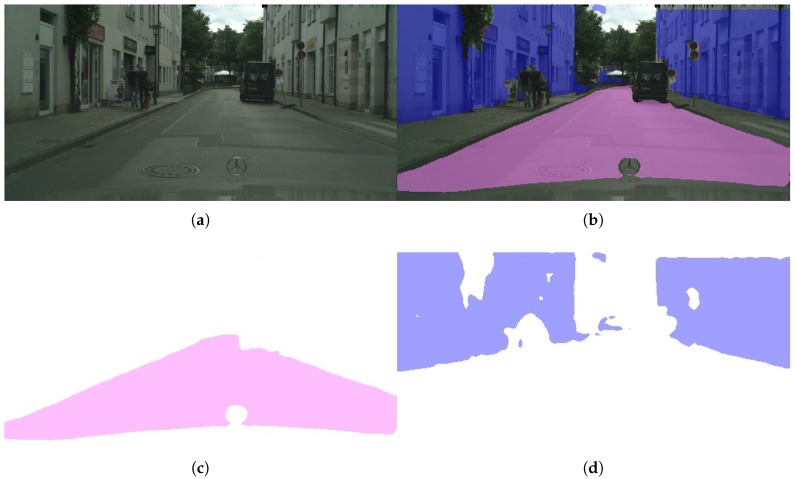
Semantic segmentation step in our method’s pipeline. (**a**) Original image. (**b**) Segmented image. (**c**) Road mask. (**d**) Fence mask.

**Figure 7 sensors-19-03224-f007:**
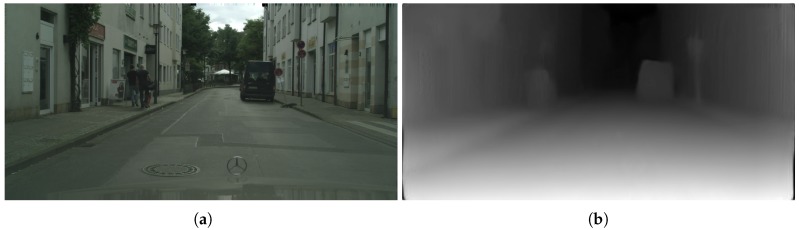
Disparity map obtained from a single image using monodepth [25]. (**a**) Original image. (**b**) Disparity map.

**Figure 8 sensors-19-03224-f008:**
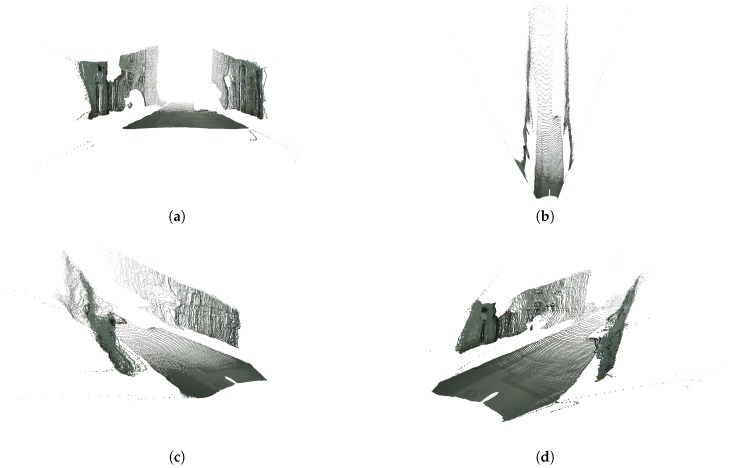
3D point clouds featuring points belonging to classes road and fence. (**a**) Front view. (**b**) Top view. (**c**) Left view. (**d**) Right view.

**Figure 9 sensors-19-03224-f009:**
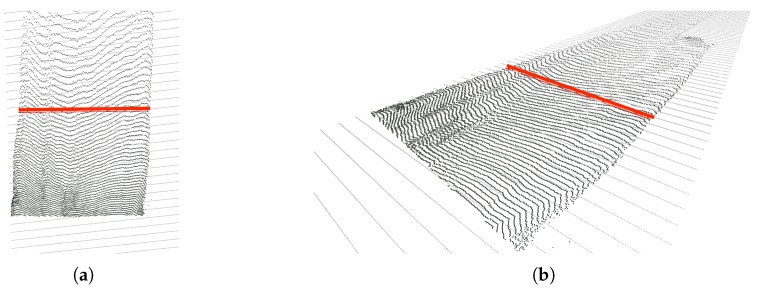
Exemplary computation of the road’s width. (**a**) Top view. (**b**) Right view.

**Figure 10 sensors-19-03224-f010:**
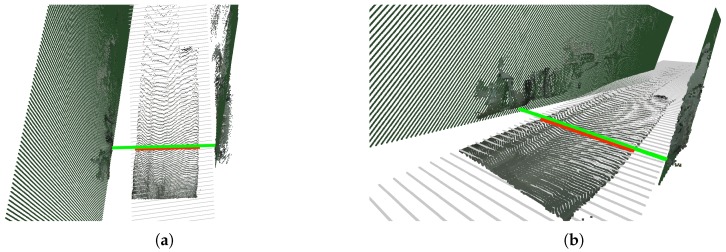
Exemplary computation of the fence-to-fence distance. The light green line represents the fence-to-fence distance, while the red line represents the road’s width. (**a**) Top view. (**b**) Right view.

**Figure 11 sensors-19-03224-f011:**
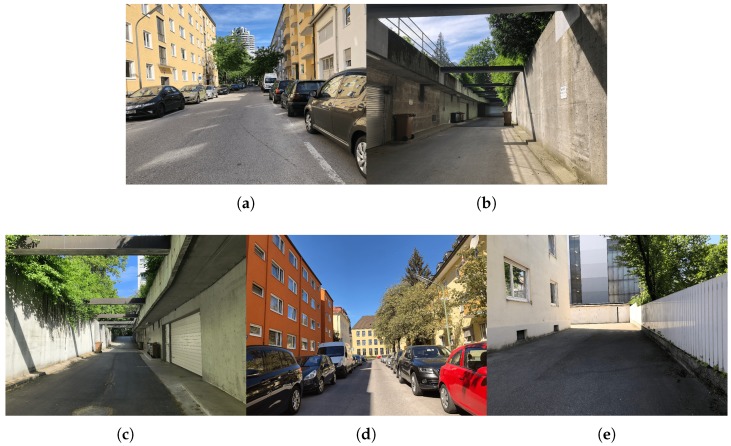
Munich test set, on which we quantitatively tested the performance of the proposed system. (**a**) Munich-1. (**b**) Munich-2. (**c**) Munich-3. (**d**) Munich-4. (**e**) Munich-5.

**Figure 12 sensors-19-03224-f012:**
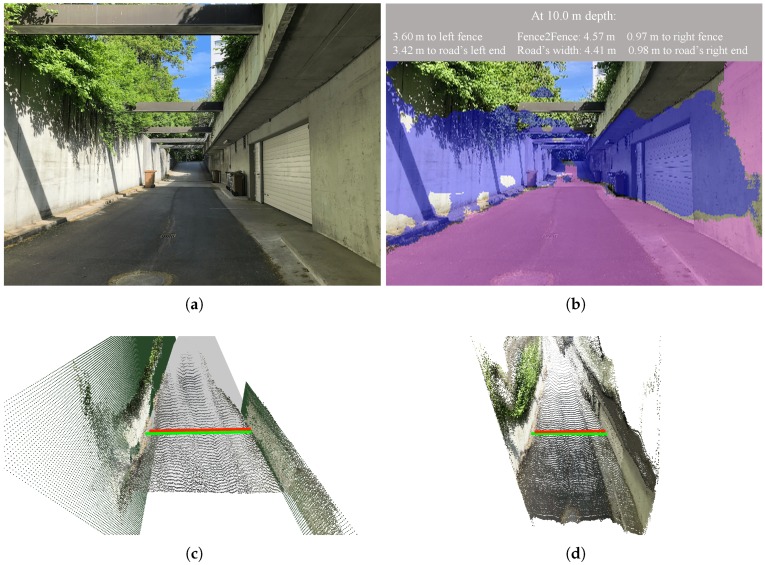
Exemplary results of applying our method on frame Munich-3 from the Munich test set. (**a**) Original image; (**b**) Output image displaying the predicted distances on the segmented frame; (**c**) Post-processed 3D point cloud, featuring the fitted planes and the computed distances: road’s width (red line) and fence-to-fence (green line); (**d**) Initial 3D point cloud, featuring the computed distances: road’s width (red line) and fence-to-fence (green line).

**Figure 13 sensors-19-03224-f013:**
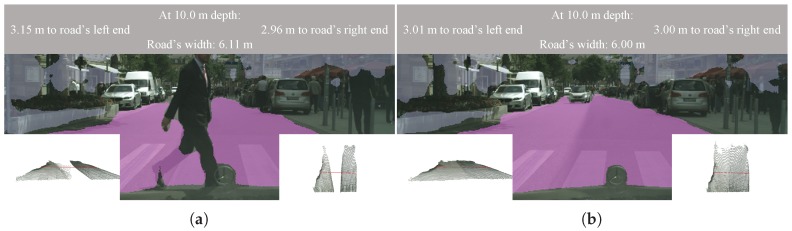
We test our system on the Stuttgart sequence of the Cityscapes dataset. The complete sequence can be found at https://youtu.be/0yBb6kJ3mgQ, as well as in the Appendix A. (**a**) Sample Frame 5185 of the Stuttgart sequence. (**b**) Sample Frame 5209 of the Stuttgart sequence.

**Table 1 sensors-19-03224-t001:** Distribution of images in the roborace750 dataset.

Set	Roborace750
training set	500 (250 (Montreal) + 250 (New York))
validation set	125 (Berlin)
test set	125 (Berlin)

**Table 2 sensors-19-03224-t002:** Results of our proposed method on the Munich test set 10 m ahead of the camera.

**Focal Length = 380 Pixels**						
**Frame**	**e (m)**	**gt Width (m)**	**rw (m)**	**f2f (m)**	**AE rw (m)**	**AE f2f (m)**
Munich-1	∼0	5.3	6.24	6.81	0.94	1.51
Munich-5	∼0	4.6	4.73	5.51	0.13	0.91
**Focal Length = 580 Pixels**						
**Frame**	**e (m)**	**gt Width (m)**	**rw (m)**	**f2f (m)**	**AE rw (m)**	**AE f2f (m)**
Munich-2	∼0	4.4	4.42	4.04	0.02	0.36
Munich-3	∼0	5.4	4.40	4.57	0.99	0.82
Munich-4	∼0	3.1	2.74	4.07	0.35	0.97
**Mean Absolute Error**						
MAE (m)	-	-	-	-	0.48	0.91

**Table 3 sensors-19-03224-t003:** Inference times (in seconds) of our proposed approach on the Munich test set.

Frame	t_read	t_seg	t_disp	t_to3D	t_road	t_rw	t_fences	t_f2f	t_total
1	0.3506	0.1521	0.0233	0.0106	0.1053	0.0013	0.0099	0.0016	0.6551
2	0.3605	0.1498	0.0233	0.0115	0.0952	0.0012	0.0144	0.0014	0.6577
3	0.3252	0.1688	0.0243	0.0110	0.0804	0.0012	0.0113	0.0016	0.6240
4	0.3551	0.1480	0.0221	0.0105	0.0338	0.0012	0.0240	0.0015	0.5965
5	0.3528	0.1456	0.0223	0.0094	0.1080	0.0013	0.0134	0.0014	0.6544
mean	0.3488	0.1528	0.0230	0.0106	0.0845	0.0012	0.0146	0.0015	0.6375
%	54.71%	23.97%	3.61%	1.7%	13.26%	0.19%	2.29%	0.24%	-

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
