# Peer review of "SemanticDepth: Fusing Semantic Segmentation and Monocular Depth Estimation for Enabling Autonomous Driving in Roads without Lane Lines"

_sensors, 2019, doi:10.3390/s19143224_

Round 1
Reviewer 1 Report
General Comments:
The work is technically sound and the paper well-written. Although the particular methodology is only applicable to a fairly narrow scope of autonomous navigation problems, the work nevertheless constitutes a meaningful incremental step in this area.
Specific Comments:
· The wording of the sentence starting on line 53 is unclear.
· Line 263: “existing noise in it” should be “existing noise” (remove “in it”)
· Line 286: “in most cases we deal with” should be “in most cases” (remove “we deal with”)
· Line 294: “we do not only” should be “we not only” (remove “do”)
· Line 388: “but also is the model” should be “but the model is also”
· Line 410: “not 100%-perfect” should be “imperfect”
Author Response
Dear reviewer, first of all we would like to thank you for the valuable comments received. We are fully convinced that these comments made it possible to substantially improve the previous version of the manuscript.
Please, find attached our answers.

Reviewer 2 Report
This paper presents a system that fuses together semantic segmentation and monocular depth estimation to robustly compute the width of the road at a certain distance on the planned trajectory. The authors use the existing FCN-8s architecture and unsupervised monocular depth estimation network to perform semantic segmentation and depth estimation, respectively. To evaluate the performance of the proposed system, the authors tested the system on Cityscapes dataset and Munich test set. Moreover, they also make their software open source. Although the authors completed many engineering works to provide solid experimental results, the contribution of this paper is unclear. For example, there are several similar papers already published online, e.g.,
Xiao Lin, et al. “Depth estimation and semantic segmentation from a single RGB image using a hybrid convolutional neural network,” Sensors, 2019.
Caner Hazirbas, et al. “FuseNet: Incorporating depth into semantic segmentation via fusion-based CNN architecture,” ACCV 2016.
Zama Ramirez, et al. “Geometry meets semantic for semi-supervised monocular depth estimation,” ACCV 2018.
I think the authors should give a discussion to clarify the difference between their work and the other published works. Moreover, it is better to compare the proposed hybrid system to the other existing methods to highlight the contribution of the proposed method.
Author Response

(The authors gave the same response as above.)

Reviewer 3 Report
The paper proposes a method of scene reconstruction from monocular images that combines semantic segmentation and depth estimation by means of deep learning. The proposed method seems to be quite specific to the environment being considered - the Roborace competition. Unfortunately, the paper fails to show how the results can be generalized to real-life automotive/urban scenarios. Also the description of the training procedure is unclear, with many references to manual tuning and 'tweaking'. The description of the SemanticDepth processing architecture is insufficient, Fig. 3 is not enough to understand how the semantic segmentation helps to estimate correct depth. To me these parts are quite independent, and their 'colaboration' is highly specific to the very particular environment considered in the paper.
Therefore, I recommend to reject the paper in its present form.
Author Response

(The authors gave the same response as above.)

Reviewer 4 Report
The manuscript introduces the system (the SemanticDepth pipeline
approach) shown in Fig. 3 for enhancing the situational awareness of
self-driving cars. Though the system development is logical, it seems
like a combination of various existing techniques. As a result, its
contribution cannot be stated as "new". Moreover, the manuscript is lack
of any theoretical development and implications. It reads like a
technical report rather than a Journal Article. The Introduction section
is not comprehensive. Though eqn. (1) is commonly used in the
literature, at least one of its relevant references should be indicated.
Eqn. (3) should be given a reference if it is not derived by the
authors. In the experiments, the performance of the SemanticDepth
should compare with that of other methods (e.g., [28]). It is not
convincing that 250 training data+50 val set+125 test set, and 500
training data+125 val. set+125 test set are sufficient for Roborace425
and Roborace725 cases, respectively.
Author Response

(The authors gave the same response as above.)

Round 2
Reviewer 2 Report
The authors already clarify the contribution of their work compared with the existing papers, and I have no further questions.
Reviewer 3 Report
The paper has been improved and now more clearly describes the approach and the aim - estimation of the road parameters rather than improvements to scene segmentation or depth estimation. I still have some reservations as to the significance of this topic in the context of general ADAS and self-driving applications, but the paper can be published in Sensors in the new version.
Reviewer 4 Report
The manuscript has been improved. It proposes a method combining various existing techniques for handling the situational awareness in an autonomous vehicle. But no theoretical development and derivation is the biggest weakness of this manuscript.